# TwinS: Revisiting Non-Stationarity in Multivariate Time Series Forecasting

## Abstract

Recently, multivariate time series forecasting tasks have garnered increasing attention due to their significant practical applications, leading to the emergence of various deep forecasting models. However, real-world time series exhibit pronounced non-stationary distribution characteristics. These characteristics are not solely limited to time-varying statistical properties highlighted by non-stationary Transformer but also encompass three key aspects: nested periodicity, absence of periodic distributions, and hysteresis among time variables. In this paper, we begin by validating this theory through wavelet analysis and propose the Transformer-based TwinS model, which consists of three modules to address the non-stationary periodic distributions: Wavelet Convolution, Period-Aware Attention, and Channel-Temporal Mixed MLP. Specifically, The Wavelet Convolution models nested periods by scaling the convolution kernel size like wavelet transform. The Period-Aware Attention guides attention computation by generating period relevance scores through a convolutional sub-network. The Channel-Temporal Mixed MLP captures the overall relationships between time series through channel-time mixing learning. TwinS achieves SOTA performance on eight datasets compared to mainstream TS models, with a maximum improvement in MSE of 25.8% over PatchTST. We have open-sourced our code to facilitate reproducibility for future research: https://anonymous.4open.science/r/TwinS-BBA3/.

## 1 Introduction

Multivariate time series forecasting (MTSF) has gained widespread prominence in real-world applications, such as weather prediction, financial risk assessment, and traffic forecasting. Transformers (Vaswani et al., 2017) have emerged as the most popular approach for this task, primarily attributed to their power in capturing temporal dependencies Wen et al. (2023). Recent advances (Wu et al., 2021; Liu et al., 2021a; Zhou et al., 2022; Nie et al., 2023) have further bolstered the popularity.

A long-lasting challenge in the realm of MTSF lies in effectively mitigating the *non-stationarity* inherent in real-world time series. In general, non-stationary time series exhibits a persistent alteration in its statistical attributes (e.g., mean and variance) and joint distribution across time, thereby diminishing its predictability. In previous work, several models have utilized time series pre-processing techniques (Passalis et al., 2019; Kim et al., 2021) to achieve stationarity or involved statistical guidance during model training (Liu et al., 2022b), resulting in significant performance enhancements.

Though promising, the above endeavors still fall short of modeling the non-stationary period distribution. To verify this point, we empirically leverage the Morlet wavelet transform on the Weather dataset (Wu et al., 2021), leading to the energy distribution in Fig 1. We observe that (i) Non-stationary time series comprises multiple nested and overlapping periods, with diverse periodic patterns and varying strengths at each time step. (ii) Non-stationary time series exhibit distinct periodic patterns segmented, indicating that a particular occurrence may only happen during specific stages or time intervals. For instance, in the left plot, the information exhibiting a periodicity ranging from 4 to 8 is exclusively observed within the time range of 180 to 330. (iii) Within time series, there are similarities in the period components but significant hysteresis in periodic distribution.

For challenge (i), although existing methods have attempted to model time series features from multiple scales using techniques such as sequence splitting (Liu et al., 2022a), downsampling (Wang et al., 2022), tree structures (Liu et al., 2021a), and pyramid architecture (Zhang et al., 2023a), they only decouple the time series information in the temporal domain and are still unable to effectively

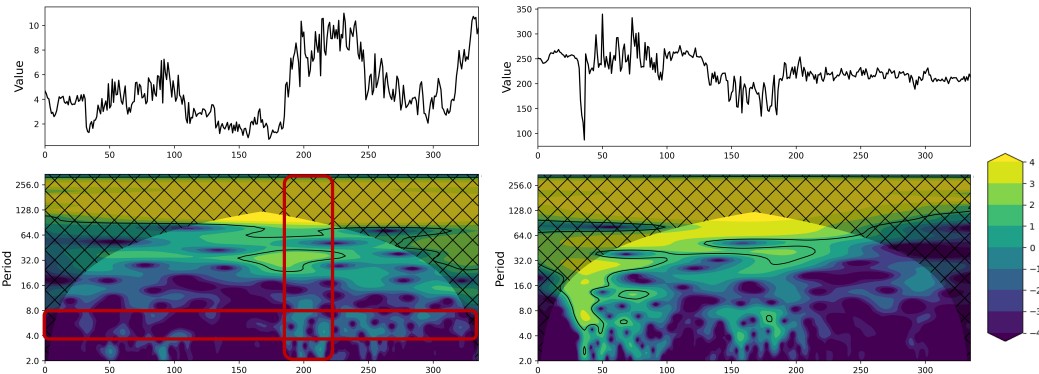

Figure 1: Examples of the wv and wd variables in the Weather dataset. The x-axis is the shared time steps. The graph above shows the variable values over time. The one below is the wavelet level plot, representing the strength of the signal's energy at different time-frequency scales.

decouple the nested periodic information in the frequency domain. For challenge (ii), traditional attention mechanisms primarily rely on explicitly modeling period information through the values of each time step, which can easily lead to the model incorrectly aggregating noise data into the temporal representation when modeling the multi-periodic non-stationary distribution. For challenge (iii), both existing channel-dependent (Wu et al., 2022; Wang et al., 2022) or independent (Zhang & Yan, 2022; Nie et al., 2023) models neglect the hysteresis among different time series. So, designing a model that can decouple *nested periods*, model *missing states* of periodicity, and capture *interconnections with hysteresis* among time series are the keys to further improving the performance of transformers in the MTSF task.

In this paper, we propose the TwinS, which comprises three key modules to address these challenges: The Wavelet Convolution Module simulates wavelet transform techniques to extract information from multiple nested periods during the initial embedding process; The Periodic Aware Attention Module, incorporates a convolution-based scoring sub-network. Building upon the wavelet convolution embedding, this mechanism effectively models non-stationary periodic distributions at various window scales. It enables TwinS to identify and understand the complex, non-stationary periodic patterns inherent in the time series data; The Channel-Temporal Mixer Module treats the time series as a holistic entity and employs a Multi-Layer Perceptron to capture overall correlations among time variables. The contributions lie in three folds:

- Upon revisiting the MTSF task, we recognized that the critical factor for improving the performance of transformer models lies in addressing nested periodicity, modeling missing states in non-stationary periodic distributions, and capturing interrelationships with hysteresis among MTS;

- We propose TwinS, a novel approach that incorporates Wavelet Convolution, Periodic Aware Attention, and Temporal-Channel Mixer MLP to model nonstationary period distribution;

- TwinS consistently outperforms eight mainstream models on eight popular datasets, and demonstrates the efficacy of each module within the framework;

## 2  RELATED WORK

**Deep Times Series Models for Periodic Learning:** Effectively modeling the underlying periodic information is crucial in MTSF tasks. Traditional transformer-based models (Zerveas et al., 2021) employ self-attention mechanisms to capture periodicity. TimesNet (Wu et al., 2022) transforms 1D time series into 2D representations using CNN to extract periodic features. Models like Prayformer (Liu et al., 2021a), Informer (Zhou et al., 2021), MICN (Wang et al., 2022), and Scaleformer (Shabani et al., 2022) extract periodic features at different scales. Autoformer (Wu et al., 2021) introduces the Auto-Correlation Mechanism for frequency domain periodic modeling. FEDformer (Zhou et al., 2022) enhances frequency domain attention with the Fourier neural operator. Non-stationary transformer (Liu et al., 2022b) addresses period distribution shift issues, and models like

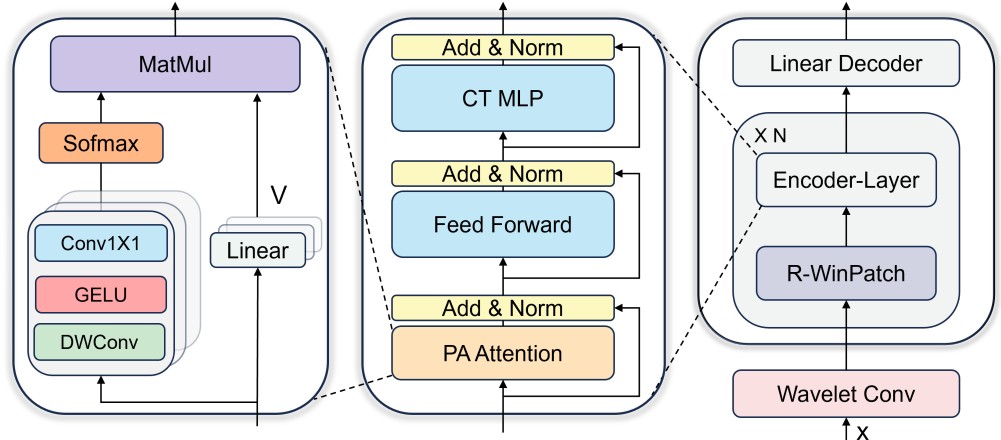

Figure 2: Overall architecture of TwinS with detailed structure of Periodic Aware Attention and Encoder-Layer.

Dlinear (Zeng et al., 2023) use linear layers for effective periodic modeling. Unlike previous methods, TwinS takes a novel perspective, starting from wavelet analysis to model the non-stationary periodic distribution in time series.

**Non-stationarity in Time Series data:** Real-world time series are often non-stationary, posing significant challenges for MTSF tasks. Traditional statistical methods like ARIMA (Box et al., 2015) use differencing to stabilize time series, while in deep models, stabilization techniques are widely employed for data preprocessing. DAIN (Passalis et al., 2019) uses non-linear neural networks and observed training distribution to stabilize time series. Revin (Kim et al., 2021) introduces trainable stabilization and destabilization layers. Non-stationary transformer (Liu et al., 2022b) incorporates De-stationary Attention. However, these methods focus on non-stationarity caused by evolving statistical features and overlook the non-stationary periodic distribution issues. Our model presents the first attempt to address these issues explicitly.

**Channel-Dependent (CD) and Channel-Independent (CI) Models:** In MTSF tasks, two commonly employed strategies are channel-dependent and channel-independent. The CD strategy, as highlighted in the works of (Wu et al., 2022; Zhou et al., 2022; 2021), involves constructing a model that predicts future values for each series by leveraging historical information from all the series. Conversely, as employed in the studies of (Zhang & Yan, 2022; Nie et al., 2023), the CI strategy employs separate modeling techniques to capture temporal dependencies for each individual series. However, the CD strategy often faces challenges such as prediction distribution bias (Han et al., 2023) and variations in the distributions of variables. This makes the CI strategy generally more robust and outperforms the CD strategy. Our model, TwinS, falls into the CI strategy category and possesses the capability to learn the relationships between time series.

## 3 METHODOLOGY OF THE TWINS

In this paper, we concentrate on long-term MTS forecasting tasks. Let $\mathbf{x}_t \in \mathbb{R}^C$ denote the value of $C$ series at time step $t$. Given a historical MTS instance $\mathbf{X}_t = [\mathbf{x}_t, \mathbf{x}_{t+1}, \cdots, \mathbf{x}_{t+L-1}] \in \mathbb{R}^{C \times L}$ with lookback window L, the objective is to predict the next T steps $\mathbf{Y}_t = [\mathbf{x}_{t+L}, \cdots, \mathbf{x}_{t+L+T-1}] \in \mathbb{R}^{C \times T}$. The aim is to learn a mapping $f(\cdot) : X_t \to Y_t$ using the proposed model.

As depicted in Figure 2, TwinS utilizes wavelet convolution for multi-period embedding. Before each encoder layer, it applies reversible window patching to capture periodicity gaps across different window scales. Within each encoder layer, it performs Periodic Aware Attention calculation, Feed-forward network, and Channel-Temporal Mixer MLP with residual structures. The Periodic Aware Attention calculation entails a multi-head sub-network to generate attention score matrices.

### 3.1 WAVELET CONVOLUTION EMBEDDING

Recent TSformers usually adopt the patch embedding method mentioned in PatchTST (Nie et al., 2023) which addresses the lack of semantic significance in individual time points while simultaneously reducing time complexity. However, this approach raises three concerns. Firstly, this method

does not effectively address the issue of nested periods in the time series. Secondly, important semantic information may become fragmented across different patches. Lastly, the predetermined patch length and stride used in the embedding process are irreversible in subsequent modeling.

In this paper, we contemplate emulating the wavelet transform methodology by embedding the time series at distinct frequency and time scales to address the initial concern. As for the subsequent concerns, we will elucidate them in Section 3.2. Specifically, for the wavelet transform (WT), we can express it in the following form:

$$WT(a, \tau, t) = \frac{1}{\sqrt{a}} \int_{-\infty}^{\infty} f(t) \cdot \psi\left(\frac{t - \tau}{a}\right) dt, \tag{1}$$

where $\psi$ denotes the wavelet basis function, $a$ represents the scale parameter responsible for scaling the wavelet basis functions, and $\tau$ represents the translation parameter governing the movement of the wavelet basis functions. The parameter $a$ can be interpreted as capturing different frequency-domain information, while $\tau$ corresponds to variations in the time domain.

Standard convolutional neural networks can be essentially regarded as discrete Gabor transforms; they perform windowed Fourier transforms in the time domain on input features:

$$GT(n, \tau, t) = \int_{-\infty}^{+\infty} f(t) \cdot g(t - \tau) \cdot e^{int} dt, \tag{2}$$

$$Conv(c, k, x) = \sum_{j=1}^{c} \sum_{p_k \in \mathcal{R}} x(p_k) \cdot \mathbf{W}_j(p_k), \tag{3}$$

where Eqn 2 is the Gabor transform, Eqn 3 is the CNN. $n$ is the number of frequency coefficients, $\tau$ is the translation parameter, $c$ is the number of CNN channels, $k$ is the kernel size and $p_k \in \mathcal{R}$ represent all the sampled points in windowed kernel size. $g(\cdot)$ is the Gabor function to scale the basis function in window size and $\mathbf{W}_j$ is the kernel weight of channel $j$. The difference between the two lies in the fact that the $g$ function is typically a Gaussian function, while in CNNs, $\mathbf{W}_j$ represents trainable weights that are automatically updated through backpropagation.

Comparing Eqn 1 and Eqn 2, the key difference between Wavelet and Gabor transform lies in the scaling factor $a$. This factor allows for a variable window in the Gabor transform. Inspired by this, we propose Wavelet Convolution as shown in Fig 3. Specifically, we apply scaling transformations to the kernel size of the convolutional kernel to correspond to the scaling transformations of wavelet basis functions. Following the setup of discrete wavelet decomposition, we exponentially modify the size of the convolutional kernel by power of 2 and subtract 1 to ensure it remains an odd number. These different scales of the convolutional kernel share the same set of parameters $\mathbf{W}$ (The parameters of smaller frequency-scaled kernels are centered around the parameters of the larger scaled kernel), resembling the concept that wavelet functions in the wavelet transform are derived from the same base function. So we can define the Wavelet Conv by Eqn 3 and $\mathbf{W}$:

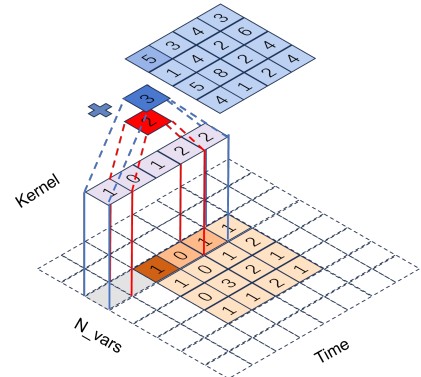

Figure 3: Wavelet Conv

$$WConv(c, k, x) = \sum_{j=1}^{c} \sum_{\mathbf{W}_{ij} \in \mathbf{W}} \sum_{p_k \in \mathcal{R}_i} x(p_k) \cdot \mathbf{W}_{ij}(p_k), \tag{4}$$

where $p_k \in \mathcal{R}_i$ is the sampled points for the kernel in $i$th frequency scale and $j$th channel (we imply time-scale transformation through the sliding of convolutions). This approach effectively captures small-scale periodic information nested within larger periods in a time series and utilizes additive concatenation to store them. It is worth mentioning that recent models (Zhang et al., 2022) have introduced trend decomposition methods, where the trend component of a time series is separately modeled using linear layers. While Wavelet convolution incorporates both information across different frequency scales and the overall trend information.

Then, given the input MTS data $\mathbf{X} \in \mathbb{R}^{1 \times C \times L}$, we employ Wavelet Convolution to get the feature map of point embedding $\mathbf{X}_{point} \in \mathbf{R}^{d \times C \times L}$ with dim $d$. Additionally, we incorporate a 1D trainable position embedding $\mathbf{E}_{pos} \in \mathbf{R}^{d \times C \times L}$:

$$\mathbf{X}_{point} = WConv(\mathbf{X}) + \mathbf{E}_{pos}. \tag{5}$$

## 3.2 PERIODIC MODELING

**Reversible Window Patching:** Inspired by the window attention mechanism in Swinformer (Liu et al., 2021b), we combine its benefits with PatchTST to achieve a refined approach. Initially, we employ point embedding by Wavelet Convolution and then utilize patching operations on the feature map using a specific window scale in each encoder layer, merging time steps within each window for subsequent attention calculations. As shown in Fig 1, when the window size is not excessively large, it does not disrupt the missing period information. Moreover, using a window scale makes it easier to detect missing period patterns. Notably, this operation is reversible, allowing flexibility in altering the window scale and restructuring the patch sequence at different model layers. Consequently, the model can effectively handle non-stationary periodic distributions across various scales:

$$\mathbf{X}_{patch}^{l} = \text{Transpose } (\text{Unfold } (\mathbf{X}_{point}, scale^{l}, stride^{l})), \tag{6}$$

$$\mathbf{X}_{point}^{l} = \text{Transpose } (\text{Fold } (\mathbf{X}_{patch}, scale^{l}, stride^{l})), \tag{7}$$

where $\mathbf{X}_{patch}^{l} \in \mathbf{R}^{C \times P^{l} \times D^{l}}$ represent the patched feature map in $l$th layer, $D^{l}$ is merged by $d \times scale^{l}$, $scale^{l}$ is equal to $stride^{l}$ to make the patches non-overlapped. It is worth mentioning that, to avoid additional complexity caused by point embeddings, we ensure the patch dimension $D$ in the initial layer remains consistent with the size in other TSformers.

Besides, we introduced intra-layer window rotation operations on $P$ dimension with size $r$, which preserve overall periodicity while improving the model's ability to resist outliers:

$$\mathbf{X}_{patch}^{l} = \text{Roll } (\mathbf{X}_{patch}, \text{shift} = r, \dim = P). \tag{8}$$

**Periods Detection Attention:** We first revisit the traditional attention mechanism in TS Transformers. Considering the difference between CI and CD models, we omit the channel dimension $C$ and represent the input sequence length as $N$. Taking a feature map $x \in \mathbf{R}^{N \times D}$, a multi-head self-attention (MHSA) block with $M$ heads is formulated as:

$$q = x\mathbf{W}_q, k = x\mathbf{W}_k, v = x\mathbf{W}_v, \tag{9}$$

$$\hat{x} = \mathbf{W}_o \cdot \text{Concat}[\sum_{m=1}^{M} \sigma(\frac{q^{(m)} \cdot k^{(m)T}}{\sqrt{D/M}}) \cdot v^{(m)}], \tag{10}$$

where all $\mathbf{W} \in \mathbf{R}^{D \times D}$ are projection matrices, $\sigma$ is the activation function. MHSA projects input feature maps onto diverse subspaces and leverages dot-product similarity to enable weighted information allocation. However, as mentioned in the Introduction, time series exist multiple non-stationary periods. For instance, Fig 4 illustrates different missing period pat-

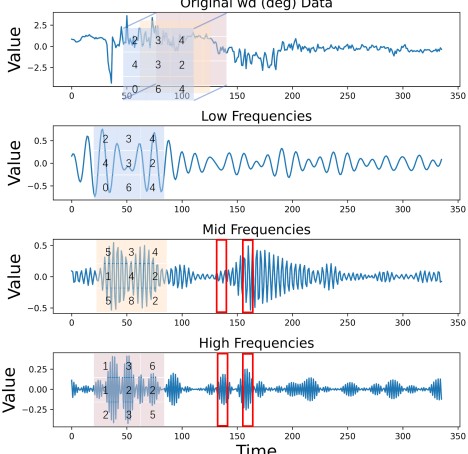

Figure 4: Frequency component figure with multiple channel Conv to aware missing periodic information

terns at different frequency scales of the wd variable in the Weather dataset. In this case, the information around time step 160 in low-frequency periodic information, should have a higher attention score to the information around time step 140. Conversely, in mid-frequency information, time step 140 may exhibit a period of absence, resulting in a lower attention score. Traditional attention mechanisms, due to the shared projection layers for each time position and dot product similarity, struggle to perceive such non-stationary periodic distributions.

Analogy to deformable convolution (Dai et al., 2017) and deformable attention (Zhu et al., 2020; Xia et al., 2022) that utilizes a sub-network to adaptively adjust the receptive field shape by fine-grained feature mapping, we employ a Convolution sub-network to aware periodicity absence with

their translation invariance, thereby guiding the information allocation in attention computation. Specifically, as shown in Fig 4, we follow the principle of multi-head attention mechanism to employ multi-head Periodic Aware sub-network to generate multiple periodic score matrices, and we enable each channel of the Conv to independently focus on a specific periodic pattern based on multiple periodic feature map embedded by Eqn 3. Subsequently, we employ a multi-layer perceptron (MLP) to aggregate the information from multiple channels within an aware head to obtain the periodic relevance scores:

$$\mathbf{W}_{score}^{(ls)} = sigmoid\ (\mathbf{W}_p \cdot \sigma(DWConv(\mathbf{X}_{patch}^{(l)})^{(s)}), \tag{11}$$

where $\mathbf{W}_{score}^{(ls)} \in \mathbf{R}^{C \times P \times P}$ is the scores matrices in $l$th layer and $s$th head, $\mathbf{X}_{patch}^{(l)} \in \mathbf{R}^{C \times P \times D_l/S}$ is the input feature map divided by $S$ heads. The number of attention heads $M$, is an integer multiple of the number of detection heads $S$. $DWConv$ is a Depthwise Separable Convolution (Chollet, 2017) utilized to detect periodic missing states. $\sigma$ is the activation function and $\mathbf{W}_p \in \mathbf{R}^{D_l/S \times P}$ is used to aggregate the output information to obtain the scores of each patch with respect to other patches. The $sigmoid$ function restricts the scores within the range of (0, 1).

Then we align the number of detection heads and attention heads to get $\mathbf{W}_{score}^{(lm)}$ and multiply the result by Formula 10:

$$\hat{\mathbf{X}}_{patch}^l = \mathbf{W}_o \cdot \text{Concat}[\sum_{m=1}^{M} \sigma(\frac{\mathbf{W}_{score}^{(lm)} \cdot q^{(lm)} \cdot k^{(lm)T}}{\sqrt{D_l/M}}) \cdot v^{(lm)}]. \tag{12}$$

We can further simplify the attention mechanism by discarding the keys and directly using the lightweight sub-network to generate the attention matrix based on the query. The final expression of Formula 10 can be rewritten as follows:

$$\hat{\mathbf{X}}_{patch}^l = \mathbf{W}_o \cdot \text{Concat}[\sum_{m=1}^{M} \sigma(\mathbf{W}_{score}^{(lm)}) \cdot v^{(lm)}]. \tag{13}$$

In this condition, the complexity of the MHSA in PatchTST and our Periodic Aware Attention can be summarized as follows:

$$\Omega(\text{MHSA}) = \underbrace{4T/P \cdot D^2}_{\text{qkv and output projection}} + \underbrace{T^2/P^2 \cdot D}_{\text{attention computation}} + \underbrace{T^2/P^2 \cdot D}_{\text{attention plus values}}, \tag{14}$$

$$\Omega(\text{PAA}) = \underbrace{2T/P \cdot D^2}_{\text{v and output projection}} + \underbrace{(k+T/P) \cdot T/P \cdot D}_{\text{attention computation}} + \underbrace{T^2/P^2 \cdot D}_{\text{attention plus values}}, \tag{15}$$

where $k$ is the kernel size of 1D $DWConv$ in the sub-network, when $k<2D$ (in practice $K = 3 \ll D$), PAA outperforms MHSA in terms of both time complexity and space complexity. In our paper, for clarity, we named the model TwinS+ when it utilizes Eqn 12 for attention computation, and TwinS when attention matrices are directly generated by Eqn 13.

## 3.3 CHANNEL-TEMPORAL MIXER MLP

In MTSF tasks, incorporating the relationships between time variables has proven beneficial for enhancing model performance (Zhang & Yan, 2022). Several models (Yu et al., 2023; Chen et al., 2023) have proposed separate modeling of dependencies in channel and time dimensions. However, For methods similar to channel attention, their physical interpretation lies in modeling the variable relationships at each time step. In such cases, the distribution hysteresis can incorrectly model the relationship information. Based on this insight, we adopt a joint learning approach instead of isolated modeling channels and time dependencies and employ a straightforward MLP to accomplish this:

$$\hat{\mathbf{H}}_{patch}^l = \mathbf{W}_2 \cdot \sigma(\mathbf{W}_1 \cdot \mathbf{H}_{patch}^l + b_1) + b_2, \tag{16}$$

where $\mathbf{H}_{patch}^l \in \mathbf{R}^{D^l \times (CP^l)}$ is the channel-temporal mixer representation after reshape operation by $\mathbf{X}_{patch} \in \mathbf{R}^{C \times P^l \times D^l}$. $\mathbf{W}_1 \in \mathbf{R}^{D^l \times h}$ and $\mathbf{W}_2 \in \mathbf{R}^{h \times D^l}$ are learnable weights. To avoid excessive parameters, we set the hidden layer dimension $h$ to a fixed constant. $b$ is the bias and $\sigma$ is the $GELU$ activate function. $\hat{\mathbf{H}}_{patch}^l$ represents the output in the $l$th encoder layer. Finally, we follow PatchTST to employ a flatten layer with linear head to get the prediction value $\mathbf{Y}$.

Table 1: Full results for long-term forecasting task. The lookback length is 96 and the prediction horizons are {96, 192, 336, 720}. The best model is in boldface, and the second best is underlined.

| Model | | TwinS+ | | TwinS | | PatchTST | | MICN | | TimesNet | | Crossformer | | Dlinear | | Stationary | | FEDformer | | ETSformer | |
|---|---|---|---|---|---|---|---|---|---|---|---|---|---|---|---|---|---|---|---|---|---|
| Metric | | MSE | MAE | MSE | MAE | MSE | MAE | MSE | MAE | MSE | MAE | MSE | MAE | MSE | MAE | MSE | MAE | MSE | MAE | MSE | MAE |
| ETTh1 | 96 | 0.379 | 0.400 | **0.376** | **0.395** | 0.426 | 0.426 | 0.398 | 0.427 | 0.384 | 0.402 | 0.391 | 0.417 | 0.386 | 0.400 | 0.513 | 0.419 | 0.376 | 0.419 | 0.340 | 0.391 |
| | 192 | **0.421** | 0.433 | 0.422 | 0.430 | 0.469 | 0.452 | 0.430 | 0.453 | 0.436 | 0.429 | 0.449 | 0.452 | 0.437 | 0.432 | 0.534 | 0.504 | 0.420 | 0.448 | 0.430 | 0.439 |
| | 336 | 0.451 | 0.438 | 0.453 | 0.438 | 0.506 | 0.473 | **0.440** | 0.460 | 0.491 | 0.469 | 0.510 | 0.489 | 0.481 | 0.459 | 0.588 | 0.535 | 0.496 | 0.459 | 0.465 | 0.497 |
| | 720 | 0.480 | 0.475 | **0.477** | **0.473** | 0.504 | 0.495 | 0.491 | 0.509 | 0.521 | 0.500 | 0.594 | 0.567 | 0.519 | 0.516 | 0.643 | 0.616 | 0.463 | 0.474 | 0.506 | 0.507 |
| ETTh2 | 96 | 0.295 | 0.347 | **0.288** | **0.342** | 0.299 | 0.351 | 0.299 | 0.364 | 0.340 | 0.374 | 0.641 | 0.549 | 0.333 | 0.387 | 0.513 | 0.419 | 0.358 | 0.397 | 0.340 | 0.391 |
| | 192 | **0.376** | **0.394** | 0.392 | 0.401 | 0.387 | 0.398 | 0.422 | 0.441 | 0.402 | 0.414 | 0.896 | 0.656 | 0.477 | 0.476 | 0.534 | 0.504 | 0.429 | 0.439 | 0.430 | 0.439 |
| | 336 | 0.424 | 0.435 | 0.420 | 0.425 | 0.426 | 0.433 | 0.447 | 0.474 | 0.452 | 0.452 | 0.936 | 0.690 | 0.594 | 0.541 | 0.588 | 0.535 | 0.496 | 0.487 | 0.485 | 0.497 |
| | 720 | 0.432 | 0.450 | **0.429** | **0.424** | 0.430 | 0.446 | 0.442 | 0.467 | 0.462 | 0.468 | 1.390 | 0.863 | 0.831 | 0.657 | 0.643 | 0.616 | 0.463 | 0.474 | 0.500 | 0.497 |
| ETTm1 | 96 | 0.324 | 0.368 | 0.325 | 0.370 | 0.342 | 0.378 | **0.316** | **0.364** | 0.338 | 0.375 | 0.366 | 0.400 | 0.345 | 0.372 | 0.386 | 0.398 | 0.764 | 0.416 | 0.375 | 0.398 |
| | 192 | 0.365 | **0.379** | 0.367 | 0.381 | 0.372 | 0.393 | **0.363** | 0.390 | 0.371 | 0.387 | 0.396 | 0.414 | 0.380 | 0.389 | 0.459 | 0.444 | 0.426 | 0.441 | 0.408 | 0.410 |
| | 336 | 0.400 | 0.409 | **0.398** | **0.405** | 0.402 | 0.413 | 0.408 | 0.426 | 0.410 | 0.411 | 0.439 | 0.443 | 0.413 | 0.413 | 0.495 | 0.464 | 0.445 | 0.459 | 0.435 | 0.428 |
| | 720 | 0.462 | 0.451 | 0.462 | **0.448** | 0.462 | 0.449 | **0.459** | 0.464 | 0.478 | 0.450 | 0.540 | 0.509 | 0.474 | 0.453 | 0.585 | 0.516 | 0.543 | 0.490 | 0.499 | 0.462 |
| ETTm2 | 96 | **0.168** | 0.254 | 0.170 | **0.253** | 0.176 | 0.258 | 0.179 | 0.275 | 0.187 | 0.267 | 0.273 | 0.346 | 0.193 | 0.292 | 0.192 | 0.274 | 0.203 | 0.287 | 0.189 | 0.280 |
| | 192 | **0.238** | 0.295 | 0.238 | **0.288** | 0.244 | 0.304 | 0.262 | 0.326 | 0.249 | 0.309 | 0.350 | 0.421 | 0.284 | 0.362 | 0.280 | 0.444 | 0.269 | 0.328 | 0.253 | 0.319 |
| | 336 | 0.302 | 0.341 | **0.302** | **0.340** | 0.304 | 0.342 | 0.305 | 0.353 | 0.321 | 0.351 | 0.474 | 0.505 | 0.369 | 0.427 | 0.334 | 0.361 | 0.325 | 0.366 | 0.314 | 0.357 |
| | 720 | 0.407 | 0.398 | 0.401 | **0.370** | 0.408 | 0.403 | **0.389** | 0.407 | 0.497 | 0.403 | 1.347 | 0.812 | 0.554 | 0.522 | 0.417 | 0.413 | 0.421 | 0.415 | 0.414 | 0.413 |
| Exchange | 96 | **0.085** | **0.203** | 0.088 | 0.210 | 0.087 | 0.206 | 0.099 | 0.240 | 0.107 | 0.234 | 0.256 | 0.367 | 0.088 | 0.218 | 0.111 | 0.237 | 0.139 | 0.276 | 0.085 | 0.204 |
| | 192 | **0.174** | **0.295** | 0.176 | 0.304 | 0.184 | 0.309 | 0.198 | 0.354 | 0.226 | 0.344 | 0.469 | 0.508 | 0.176 | 0.315 | 0.219 | 0.335 | 0.256 | 0.369 | 0.182 | 0.303 |
| | 336 | 0.331 | **0.415** | 0.334 | 0.417 | 0.322 | 0.421 | 0.302 | 0.447 | 0.367 | 0.448 | 0.901 | 0.741 | 0.313 | 0.427 | 0.421 | 0.841 | 0.426 | 0.464 | 0.348 | 0.428 |
| | 720 | 0.864 | 0.702 | 0.859 | 0.692 | 0.825 | 0.680 | **0.738** | **0.662** | 0.964 | 0.746 | 1.398 | 0.965 | 0.839 | 0.695 | 1.092 | 0.769 | 1.090 | 0.800 | 1.025 | 0.774 |
| Weather | 96 | **0.149** | **0.203** | 0.150 | 0.207 | 0.176 | 0.218 | 0.161 | 0.229 | 0.172 | 0.220 | 0.164 | 0.232 | 0.196 | 0.255 | 0.73 | 0.223 | 0.217 | 0.296 | 0.237 | 0.312 |
| | 192 | **0.204** | **0.250** | 0.205 | 0.251 | 0.223 | 0.259 | 0.220 | 0.281 | 0.219 | 0.261 | 0.211 | 0.276 | 0.237 | 0.296 | 0.245 | 0.285 | 0.276 | 0.336 | 0.237 | 0.213 |
| | 336 | **0.262** | **0.286** | 0.262 | 0.290 | 0.277 | 0.297 | 0.278 | 0.331 | 0.280 | 0.306 | 0.269 | 0.327 | 0.283 | 0.335 | 0.321 | 0.338 | 0.339 | 0.380 | 0.298 | 0.353 |
| | 720 | 0.344 | 0.345 | 0.342 | 0.344 | 0.353 | 0.347 | **0.311** | 0.356 | 0.365 | 0.359 | 0.355 | 0.404 | 0.345 | 0.381 | 0.414 | 0.410 | 0.403 | 0.428 | 0.352 | 0.388 |
| Electricity | 96 | 0.154 | 0.263 | **0.151** | **0.260** | 0.190 | 0.296 | 0.164 | 0.269 | 0.168 | 0.272 | 0.254 | 0.347 | 0.197 | 0.282 | 0.169 | 0.273 | 0.193 | 0.308 | 0.187 | 0.304 |
| | 192 | **0.166** | **0.273** | 0.166 | 0.272 | 0.199 | 0.304 | 0.177 | 0.285 | 0.184 | 0.289 | 0.261 | 0.353 | 0.196 | 0.285 | 0.182 | 0.286 | 0.201 | 0.315 | 0.199 | 0.315 |
| | 336 | 0.197 | 0.302 | **0.195** | **0.300** | 0.217 | 0.319 | 0.198 | 0.304 | 0.198 | 0.300 | 0.273 | 0.364 | 0.209 | 0.301 | 0.200 | 0.304 | 0.214 | 0.329 | 0.212 | 0.329 |
| | 720 | 0.230 | 0.329 | 0.227 | 0.326 | 0.258 | 0.352 | 0.212 | 0.321 | **0.220** | **0.320** | 0.303 | 0.388 | 0.245 | 0.333 | 0.222 | 0.321 | 0.246 | 0.355 | 0.233 | 0.345 |
| Traffic | 96 | 0.468 | **0.315** | 0.477 | 0.324 | **0.462** | 0.315 | 0.519 | 0.309 | 0.593 | 0.321 | 0.558 | 0.320 | 0.650 | 0.396 | 0.612 | 0.338 | 0.587 | 0.366 | 0.607 | 0.392 |
| | 192 | 0.479 | 0.328 | 0.481 | 0.332 | **0.473** | 0.321 | 0.537 | **0.315** | 0.617 | 0.336 | 0.569 | 0.321 | 0.652 | 0.397 | 0.613 | 0.340 | 0.604 | 0.373 | 0.621 | 0.399 |
| | 336 | **0.477** | 0.325 | 0.482 | 0.330 | 0.494 | 0.331 | 0.534 | **0.313** | 0.629 | 0.336 | 0.591 | 0.328 | 0.605 | 0.373 | 0.618 | 0.328 | 0.621 | 0.383 | 0.622 | 0.396 |
| | 720 | 0.510 | 0.338 | **0.502** | 0.339 | 0.522 | 0.342 | 0.577 | **0.325** | 0.640 | 0.350 | 0.652 | 0.359 | 0.650 | 0.396 | 0.653 | 0.355 | 0.626 | 0.382 | 0.622 | 0.396 |
| $1^{st}$Count | | 11 | 10 | **13** | **17** | 2 | 1 | 9 | 5 | 0 | 1 | 0 | 0 | 0 | 0 | 0 | 0 | 0 | 0 | 1 | 0 |

## 4 EXPERIMENTS

In this section, we delve into the performance of TwinS and its variants on popular time series forecasting datasets and thoroughly analyze the contribution of each module in the model structure. Additionally, we provide visualizations of attention-receptive fields to validate the effectiveness of the periodicity detection attention and hyper-parameters analysis. For details of experimental settings, dataset descriptions, and baseline model specifications, please refer to Appendix A.

### 4.1 MAIN RESULT

We specifically focus on long-term forecasting tasks. As shown in Table 1, TwinS and TwinS+ consistently generally achieve SOTA performance compared to eight mainstream TS models. For datasets with more pronounced non-stationary periodic distributions, such as Electricity and Weather (as illustrated in Appendix D), CNN-based models outperform Transformer-based models. This can be attributed to the inherent nature of convolutional neural networks, which excel at handling periodic information as filters. TwinS effectively combines the advantages of both models to capture periodic patterns. Regarding the Exchange and ETTm1 datasets, TwinS and MICN demonstrate comparable performance. This can be attributed to the relatively stationary nature of these datasets, which allows TwinS to degrade into a multi-scale convolution model. As to the traffic dataset, the large number of temporal variables may limit the expressive capacity of the CT-MLP hidden layer, as it is restricted by the uniform constant setting we employed. In such cases, it is advisable to consider using recent modules (Zhang et al., 2023b) designed to model the relationships between variables as replacements for the CT-MLP.

### 4.2 ABLATION STUDY

**Modules Ablation.** To comprehensively investigate the contributions of each module within our proposed framework, we conduct meticulous ablation experiments on the ETTh1 and Weather datasets. The preliminary analysis of Tab 2 shows that TwinS consistently demonstrates the highest performance in the experiments. Each individual module plays a pivotal and constructive role in enhancing the overall performance of the framework. Moreover, we have observed a discernible correlation between the efficacy of these modules and the unique characteristics exhibited by the respective datasets, as elaborated upon in Appendix D. For instance, the weather dataset manifests substantial temporal distribution lags between variables, thereby highlighting the critical signifi-

Table 2: Model architecture ablations. "w/o" denotes removing or replacing the block with the corresponding method in PatchTST. WC: Wavelet Convolution; RWP: Reversible Window Patching; CT-MLP: Channel-Temporal Mixer MLP; PAA: Periodic Aware Attention. The results are averaged from three experimental trials.

| Model | | TwinS | | w/o WC,RWP | | w/o CT-MLP | | w/o PAA | |
|---|---|---|---|---|---|---|---|---|---|
| Metric | | MSE | MAE | MSE | MAE | MSE | MAE | MSE | MAE |
| ETTh1 | 96 | **0.376** | **0.395** | 0.388 (-3.2%) | 0.401 (-1.5%) | 0.379 (-0.8%) | 0.398 (-0.8%) | 0.384 (-2.1%) | 0.402 (-1.8%) |
| | 192 | **0.422** | **0.430** | 0.438 (-3.8%) | 0.440 (-2.4%) | 0.424 (-0.5%) | 0.431 (-0.2%) | 0.429 (-1.7%) | 0.436 (-1.4%) |
| | 336 | **0.453** | **0.438** | 0.476 (-5.1%) | 0.453 (-3.4%) | 0.458 (-1.1%) | 0.439 (-0.2%) | 0.463 (-2.2%) | 0.447 (-2.1%) |
| | 720 | 0.477 | 0.473 | 0.481 (-0.8%) | 0.477 (-0.9%) | **0.473 (+0.8%)** | **0.470 (+0.6%)** | 0.490 (-3.8%) | 0.481 (-3.8%) |
| | AVG | **0.432** | **0.434** | 0.446 (-3.5%) | 0.443 (-2.1%) | 0.434 (-0.5%) | 0.435 (-0.5%) | 0.442 (-2.3%) | 0.442 (-1.9%) |
| Weather | 96 | **0.150** | **0.207** | 0.156 (-4.0%) | 0.212 (-2.4%) | 0.160 (-6.7%) | 0.210 (-1.4%) | 0.155 (-3.3%) | 0.211 (-1.9%) |
| | 192 | **0.205** | **0.251** | 0.211 (-2.9%) | 0.254 (-1.2%) | 0.228 (-11.2%) | 0.265 (-5.6%) | 0.207 (-1.0%) | 0.254(-1.2%) |
| | 336 | **0.262** | **0.290** | 0.267 (-1.9%) | 0.295 (-1.7%) | 0.285 (-8.8%) | 0.299 (-3.1%) | 0.265 (-1.2%) | 0.293 (-1.0%) |
| | 720 | **0.342** | **0.344** | 0.349 (-2.0%) | 0.352 (-2.3%) | 0.360 (-5.3%) | 0.354 (-3.0%) | 0.344 (-0.9%) | 0.347 (-0.9%) |
| | AVG | **0.240** | **0.273** | 0.246 (-2.5%) | 0.278 (-1.8%) | 0.258 (-7.5%) | 0.282 (-3.3%) | 0.243 (-1.3%) | 0.276 (-1.1%) |

Table 3: Comparative experiments regarding embedding types, we only modify the embedding module in TwinS. For Linear-patch, we replace wavelet convolution and window patching, while for remaining methods, we only replace wavelet convolution.

| Model | | Ours | | Linear-patch | | 1x3 Conv | | FFT-image | | FFT-weight | | Wavelet-weight | | Inception | |
|---|---|---|---|---|---|---|---|---|---|---|---|---|---|---|---|
| Metric | | MSE | MAE | MSE | MAE | MSE | MAE | MSE | MAE | MSE | MAE | MSE | MAE | MSE | MAE |
| ETTm2 | 96 | **0.170** | **0.253** | 0.174 | 0.258 | 0.174 | 0.256 | 0.173 | 0.254 | 0.177 | 0.261 | 0.180 | 0.257 | 0.178 | 0.259 |
| | 192 | **0.238** | 0.288 | 0.244 | 0.296 | 0.241 | 0.290 | 0.240 | **0.282** | 0.260 | 0.322 | 0.249 | 0.302 | 0.250 | 0.308 |
| | 336 | 0.302 | 0.340 | 0.308 | **0.339** | 0.307 | 0.344 | **0.301** | 0.345 | 0.319 | 0.351 | 0.312 | 0.352 | 0.318 | 0.354 |
| | 720 | **0.401** | **0.370** | 0.410 | 0.396 | 0.405 | 0.381 | 0.401 | 0.374 | 0.424 | 0.405 | 0.408 | 0.389 | 0.419 | 0.402 |
| | AVG | **0.278** | **0.313** | 0.285 | 0.318 | 0.282 | 0.317 | 0.279 | 0.314 | 0.295 | 0.335 | 0.287 | 0.325 | 0.291 | 0.331 |
| Electricity | 96 | **0.151** | **0.260** | 0.170 | 0.281 | 0.166 | 0.281 | 0.160 | 0.268 | 0.179 | 0.287 | 0.174 | 0.283 | 0.184 | 0.291 |
| | 192 | **0.166** | **0.272** | 0.181 | 0.296 | 0.177 | 0.285 | 0.172 | 0.281 | 0.182 | 0.291 | 0.179 | 0.299 | 0.188 | 0.301 |
| | 336 | **0.195** | **0.300** | 0.220 | 0.327 | 0.206 | 0.304 | 0.198 | 0.300 | 0.211 | 0.314 | 0.230 | 0.324 | 0.225 | 0.320 |
| | 720 | 0.227 | 0.326 | 0.233 | 0.338 | 0.232 | 0.329 | **0.220** | **0.320** | 0.252 | 0.347 | 0.249 | 0.341 | 0.237 | 0.344 |
| | AVG | **0.185** | **0.290** | 0.201 | 0.311 | 0.195 | 0.300 | 0.188 | 0.292 | 0.206 | 0.310 | 0.208 | 0.312 | 0.209 | 0.314 |

cance of incorporating the CT-MLP module (notably, removing this module leads to a maximum 11.2% performance drop). Furthermore, excluding both the wavelet convolution embedding and the periodic-aware attention components yields a noticeable deterioration in performance, thus further reinforcing our initial hypothesis regarding the indispensability of considering nested and missing periods in the modeling of time series data.

**Embedding Type:** In this section, we extensively investigate various embedding methods used in recent time series models, as well as different variations of Wavelet Convolution on two datasets. The specific methods we examine are as follows:

**Linear-patch.** Appearing in PatchTST (Nie et al., 2023), which utilizes shared linear layers to embed time series with patches. **1 x 3 Conv:** Appearing in MTPNet (Zhang et al., 2023a), which utilizes a 1x3 Conv to get fine-grained feature embedding. **FFT-image:** Appearing in TimesNet (Wu et al., 2022). In our deployment, it first transforms 1D time series to 2D time image by dominant periods and uses a 3x3 Conv to embed and finally reshape to 1D. We input multiperiod data parallel to the transformer and perform an average operation before the linear head. **FFT-weight:** The variant of Wavelet Convolution utilizes Fast Fourier Transform to dynamically identify dominant periods and assign weights as the kernel scales. **Wavelet-weight:** Another variant employs Wavelet Transform to adaptively detect dominant periods and assign corresponding weights as the kernel scales. **Inception:** The Inception V1 model (Szegedy et al., 2017) is utilized, employing unshared convolution kernels to embed time series.

According to Tab 3, our embedding method demonstrates superior performance on both datasets. The closely-following method is the FFT-image embedding approach. Similar to our method, it transforms into a 2D periodic feature map (as depicted in Appendix B). The vertical dimension of the feature map signifies the same phase between different periods. By comparing the phase information, it becomes possible to detect missing periods. However, the FFT-image method incurs high computational costs due to the inability of the transformer structure to parallelize computations caused by the differing feature map sizes. Compared to linear patch embedding, the 1x3 convolution embedding followed by patching yields superior results. This suggests that applying fine-grained convolution before patching effectively mitigates the issue of information fragmentation. Although FFT-weight and Wavelet-weight are adaptive versions of wavelet convolution, their performance is unsatisfactory. This may be attributed to the incomplete selection of dominant periods in the adaptive methods, leading to the oversight of crucial periodic information. The subpar performance of the

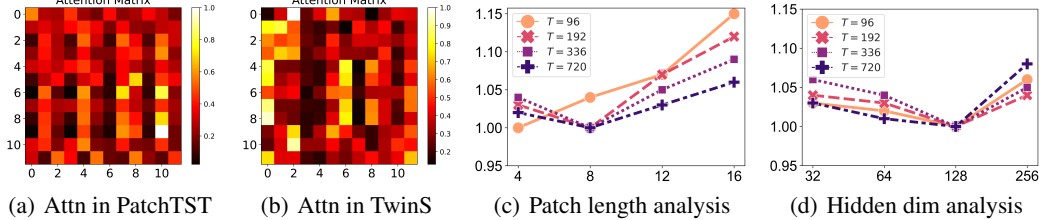

(a) Attn in PatchTST     (b) Attn in TwinS     (c) Patch length analysis     (d) Hidden dim analysis

Figure 5: Further analysis. (a) Visualization for attention matrices in PatchTST. (b) Visualization for attention matrices in TwinS. (c) MSE against the patch length on Weather dataset. (d) MSE against hidden dim of CT-MLP on ETTm2 dataset. We normalize (cd) to highlight the differences using the optimal result.

Inception model suggests that only shared convolutional kernels in Wavelet Conv can effectively extract nested periodic information.

**Attention type:** In this section, we delve into the performance evaluation of various Period Aware Attention (PAA) module designs. To emphasize the impact of this module, we substitute the conventional attention module within the Traditional TSformer (PatchTST) framework. Based on Tab 4, it is evident that utilizing a convolutional sub-network for period awareness achieves optimal performance, whereas replacing it with an MLP significantly degrades performance. This validates that convolutional neural networks, leveraging translational invariance, are better suited for this task. Furthermore, applying a pruning strategy to the attention matrix also leads to performance degra-

Table 4: Experiments regarding attention types. PAA+conv: version in TwinS+; PAA/conv: version in TwinS; PAA/mlp: use MLP to generate attention matrices; PAA/0.2: use the score matrix to drop 20% data.

| Model | **PAA+conv** | | PAA/conv | | PAA/mlp | | PAA/0.2 | |
|---|---|---|---|---|---|---|---|---|
| Metric | MSE | MAE | MSE | MAE | MSE | MAE | MSE | MAE |
| ETTh2 96 | **0.290** | **0.347** | 0.296 | 0.350 | 0.308 | 0.366 | 0.303 | 0.353 |
| ETTh2 192 | **0.385** | **0.395** | 0.388 | 0.396 | 0.402 | 0.410 | 0.394 | 0.405 |
| ETTh2 336 | 0.431 | 0.436 | 0.429 | **0.432** | 0.430 | 0.437 | 0.428 | 0.435 |
| ETTh2 720 | **0.427** | **0.439** | 0.428 | 0.441 | 0.461 | 0.478 | 0.448 | 0.464 |
| Exchange 24 | 0.085 | 0.203 | **0.084** | **0.201** | 0.090 | 0.211 | 0.090 | 0.208 |
| Exchange 36 | **0.182** | **0.308** | **0.182** | 0.309 | 0.188 | 0.314 | 0.189 | 0.313 |
| Exchange 48 | 0.320 | 0.424 | **0.319** | **0.420** | 0.330 | 0.426 | 0.328 | 0.430 |
| Exchange 60 | **0.823** | **0.677** | 0.828 | 0.681 | 0.831 | 0.692 | 0.826 | 0.680 |

dation. This could be attributed to time series prediction tasks being information-dense, and pruning this information may result in omitting crucial insights.

## 4.3 FURTHER ANALYSIS

**Attention Visualization:** We visualize the attention matrices of PatchTST and TwinS (more results are provided in Appendix C). From Fig 5(a), when the prediction horizon is 96 and the patch length is 8, it can be observed that the standard attention mechanism in PatchTST focuses on certain fixed pattern information, and many time patches have similar weights. This uniformity limits the representation of multiple non-stationary periodic features to some extent. In contrast, Fig 5(b) shows that TwinS, through the attention matrices generated by the convolutional sub-network, can better detect the distribution of non-stationary periods in the time series and adaptively generate weights.

**Effect of Hyper-parameters:** We investigate the impact of two hyper-parameters, patch length and the dimension of the CT-MLP hidden layers, on the model performance using the Weather and ETTm2 datasets. As shown in Fig 5(c), using non-overlapping patches with a length of 8 generally achieves the best performance. The patch length should be appropriately reduced for shorter prediction windows and conversely, larger patches are more suitable for longer prediction tasks. As shown in Fig 5(d), the model's performance may exhibit fluctuations depending on the hidden dim. Generally, a hidden dim of 128 can better capture the relationships between series.

## 5 CONCLUSION

This paper revisits the issue of non-stationarity in MTSF tasks, highlighting the significance of addressing nested periodic information, modeling missing periodic states, and considering series relationships with hysteresis. To tackle these challenges, we propose TwinS, which comprises three modules: Wavelet Convolution, Periodicity Aware Attention, and CT-MLP. Our model demonstrates excellent performance on eight real-world benchmark datasets and we provide detailed derivations and ablation studies to validate the effectiveness of each component within the framework. TwinS represents an initial exploration in addressing the problem of nonstationary periodic distribution. Moving forward, we aim to further investigate more efficient and interpretable methods.

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

## A  IMPLEMENT DETAILS

### A.1  DATASETS

Our experiments are carried out on Five real-world datasets as described below:

**Traffic**[1] dataset contains data from 862 sensors installed on highways in the San Francisco Bay Area, measuring road occupancy. This dataset is recorded on an hourly basis over two years, resulting in

---

[1]https://pems.dot.ca.gov/

a total of 17,544 time steps.

**Electricity**[2] dataset records the power usage of 321 customers. The data is logged hourly over a two-year period, amassing a total of 26,304 time steps.

**ETT**[3] dataset are procured from two electricity substations over two years. They provide a summary of load and oil temperature data across seven variables. For ETTm1 and ETTm2, the "m" signifies that data was recorded every 15 minutes, yielding 69,680 time steps. ETTh1 and ETTh2 represent the hourly equivalents of ETTm1 and ETTm2, each containing 17,420 time steps.

**Exchange**[4] dataset details the daily foreign exchange rates of eight countries, including Australia, British, Canada, Switzerland, China, Japan, New Zealand, and Singapore from 1990 to 2016.

**Weather**[5] dataset is a meteorological collection featuring 21 variates, gathered in the United States over a four-year span.

## A.2 BASELINES

Our baseline models include:

**PatchTST** (Zeng et al., 2023)follows the setting of the ViT (Vision Transformer) model by projecting the time series into linear patches and modeling the temporal dependencies based on these patches.

**MICN** (Wang et al., 2022) employs multi-scale sampling and utilizes convolutional neural networks to model multi-scale periodic information.

**TimesNet** (Wu et al., 2022) transform 1d time series data to 2d periodic image and use CNNs to extract the feature map.

**Crossformer** (Zhang & Yan, 2022) use two directional attention computations: Temporal attention and Channel attention, and use a router to reduce complexity

**Dlinear** (Zeng et al., 2023) proposes using a simple two-layer MLP to model periodic information.

**Stationary** (Liu et al., 2022b) propose Non-stationary Attention, normalization, and denormalization operation to tackle the issues of Non-stationary in time series data.

**FEDformer** (Zhou et al., 2022) learn the time dependency in frequency domain by FNO.

**ETSformer** (Woo et al., 2022) decomposes the time series into three parts: Level, Growth, and Seasonal and exploits the principle of exponential smoothing in improving Transformer in MTSF tasks.

## A.3 EXPERIMENTS SETTING

All experiments are conducted on the NVIDIA RTX3090-24G and A6000-48G GPUs. The Adam optimizer is chosen. The kernel size of the periodic aware sub-network is 3. A grid search is performed to determine the optimal hyperparameters, including the learning rate from {0.00005, 0.0001}, patch length from {4, 8, 12} (consistent with stride length), hidden layer dimensions of CT-MLP from {64, 128, 256, 512}, hidden patch dim from {32, 64, 128, 256}, the number of attention and periodic aware heads are from {4, 8}.

# B ABOUT FFT-IMAGE

When applying the TimesNet approach to transform a 1D time series into a 2D image, as shown in Fig B, the vertical axis represents the same phase information between different periods. Assuming the time series has a strictly stationary distribution, the information within the same phase is highly similar. Using 2D convolution, we can detect the phase differences and perceive missing periods at a finer granularity. Both TimesNet and the results in Tab 3 validate this assumption.

However, due to the structural limitations of Transformers, we can only input fixed-length feature maps to the model. If we want to employ a convolutional subnet similar to TwinS to perceive missing periods, whether by using recurrence or mapping different-sized period images to a sufficiently large area as shown in Fig B, the time complexity becomes prohibitively high. Although experimental

---

[2]https://archive.ics.uci.edu/ml/datasets/ElectricityLoadDiagrams20112014

[3]https://github.com/zhouhaoyi/ETDataset

[4]https://github.com/laiguokun/multivariate-time-series-data

[5]https://www.bgc-jena.mpg.de/wetter

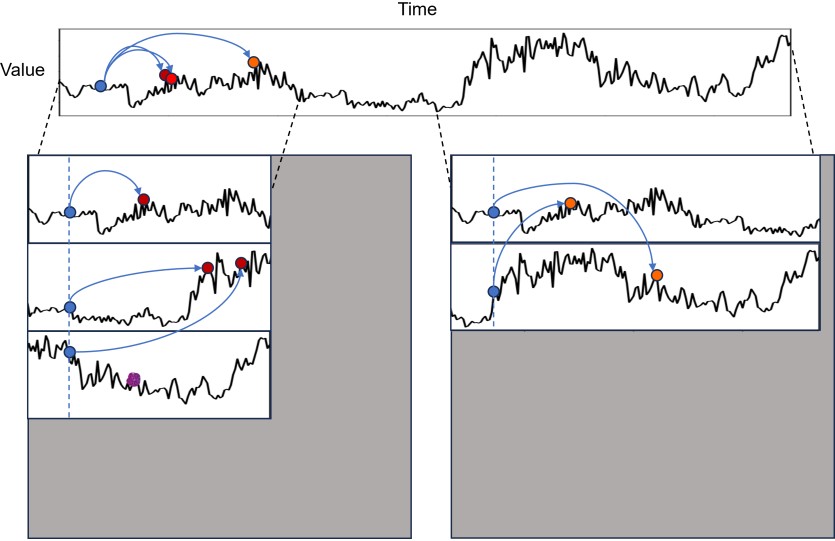

Figure 6: Examples of FFT-image

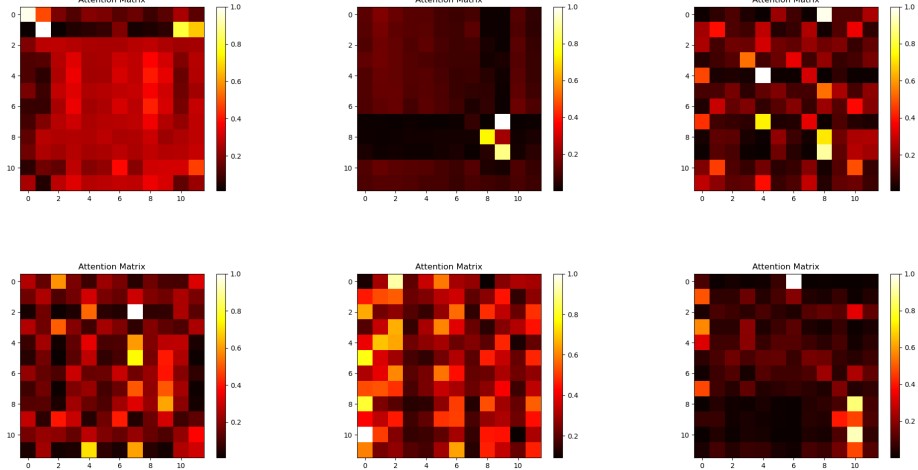

Figure 7: Attn in PatchTST

results demonstrate significant performance improvement using these methods, the computational cost is impractical. Therefore, we report the version of the model based on TwinS.

## C    ATTENTION VISUALIZATION

Here, we present heat maps of attention matrices for the first head of the first data in the batch during the testing process.

Through comparison, we can observe that traditional attention mechanisms tend to distribute information evenly or pay extra attention to specific fixed positions. In contrast, TwinS obtains attention scores directly through convolutions, allowing for more freedom and making detecting missing periods easier. Additionally, experimental results indicate that using a convolutional subnet for attention computation in TwinS leads to faster convergence than traditional methods.

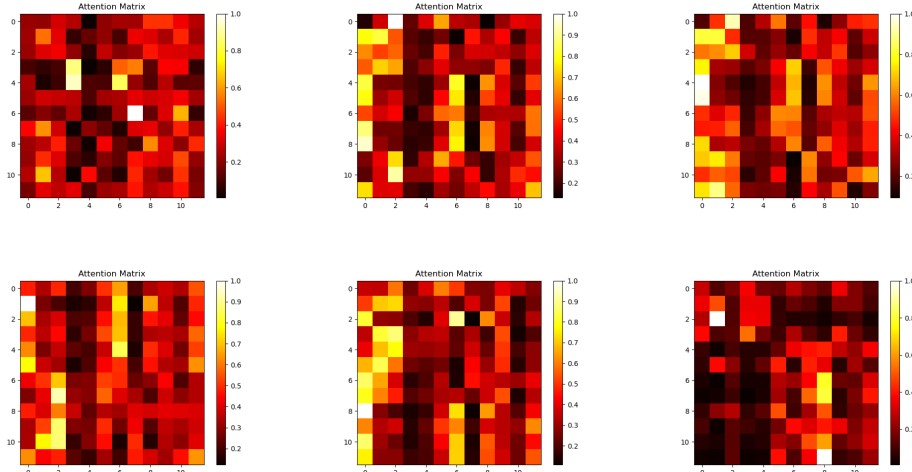

Figure 8: Attn in TwinS

## D  WAVELET ANALYSIS FIGURE

To illustrate the non-stationary periodic distributions, we selected four variables from each dataset and visualized them using wavelet analysis. Wavelet analysis helps us observe the periodic characteristics of signals at different scales and frequencies. Through this visualization, we can better understand the non-stationary periodic distributions present in the dataset.

We selected four variables with distinct features for each dataset and performed wavelet analysis on them. By applying wavelet transform to the signals, we can decompose them into components at different frequencies and observe their variations at different time scales. The results of these wavelet analyses can be presented through graphical visualizations, enabling a more intuitive observation of the periodic characteristics of the signals.

Using this visualization approach, we can identify the presence of non-stationary periodic distributions in the dataset and gain insights into the patterns of variation at different time scales. This helps us select appropriate models and methods to handle datasets with non-stationary periodic distributions.

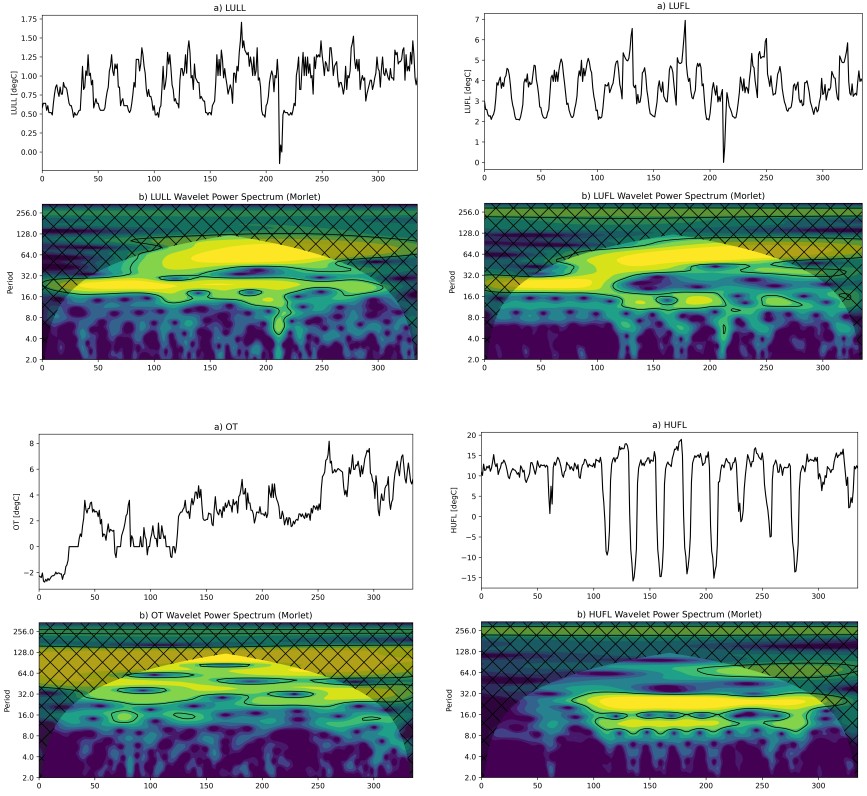

Figure 9: ETTh1 dataset

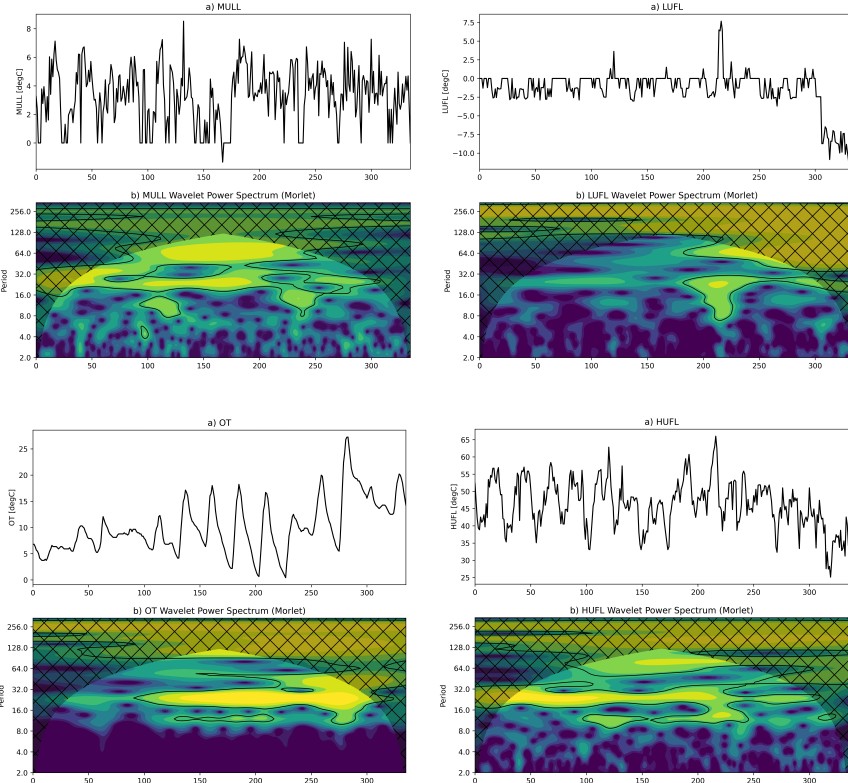

Figure 10: ETTh2 dataset

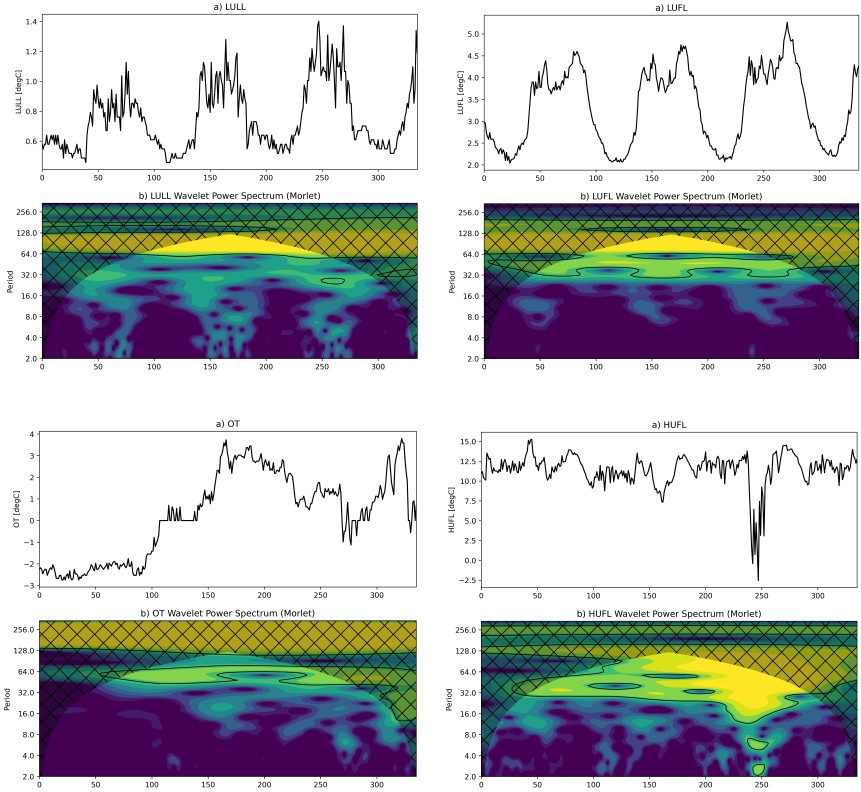

Figure 11: ETTm1 dataset

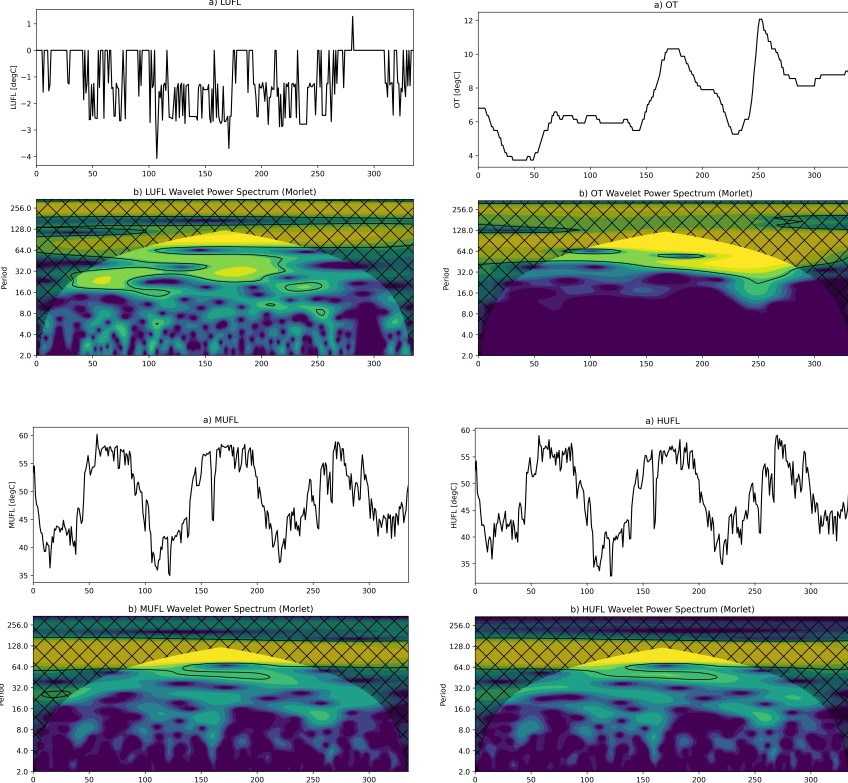

Figure 12: ETTm2 dataset

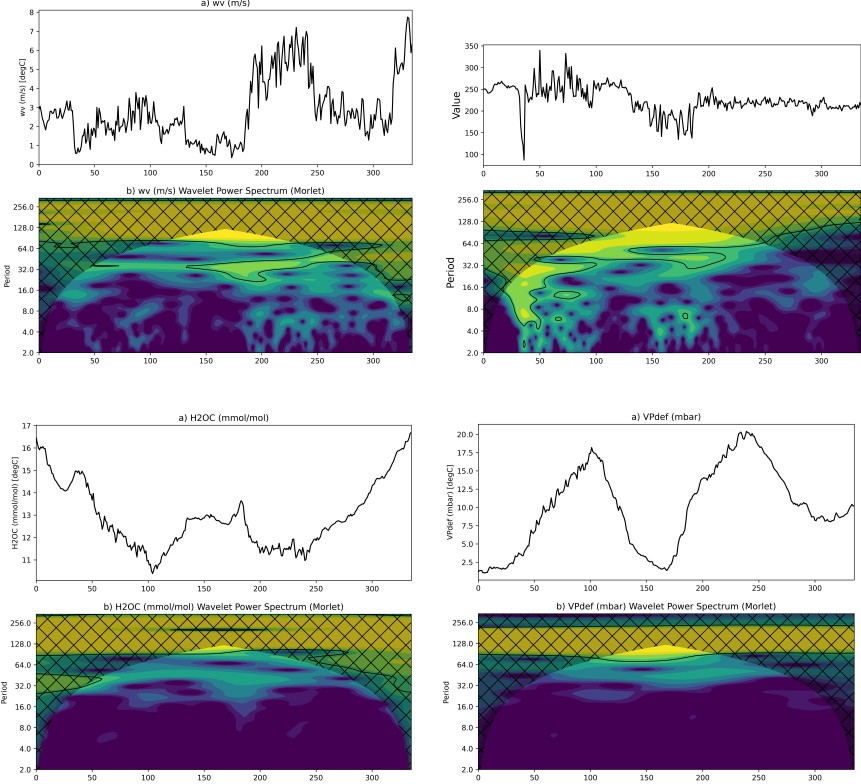

Figure 13: Weather dataset

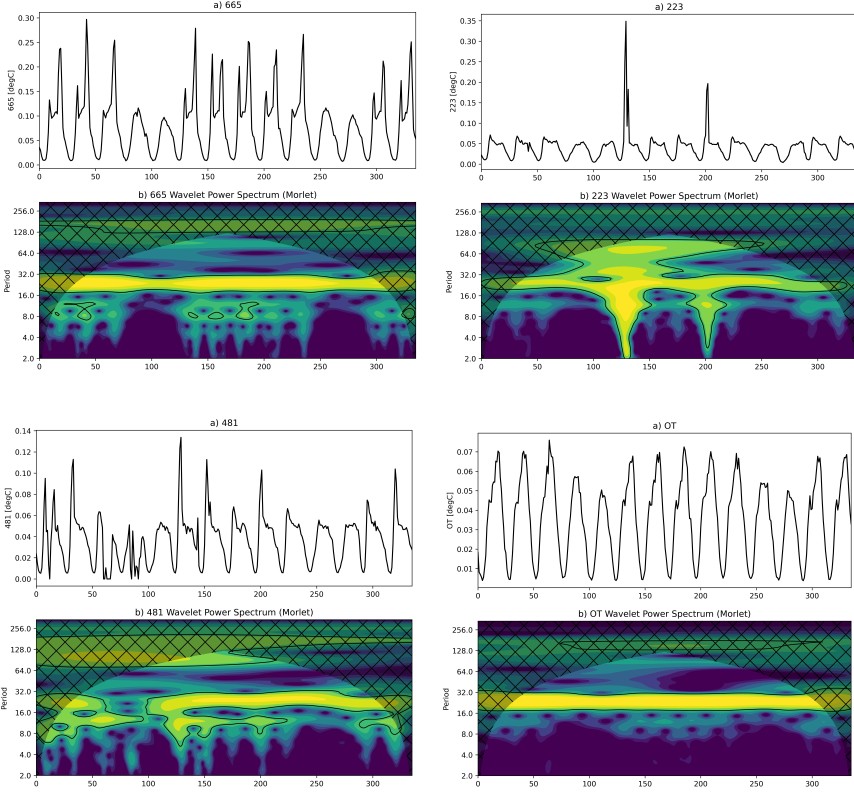

Figure 14: Traffuc dataset

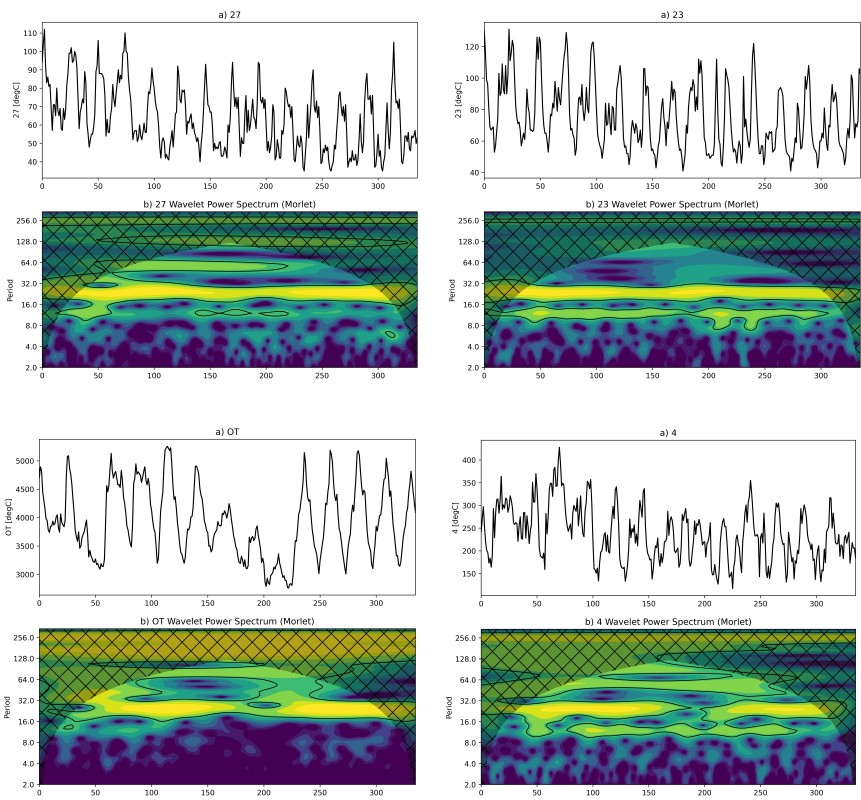

Figure 15: Eclectricity dataset

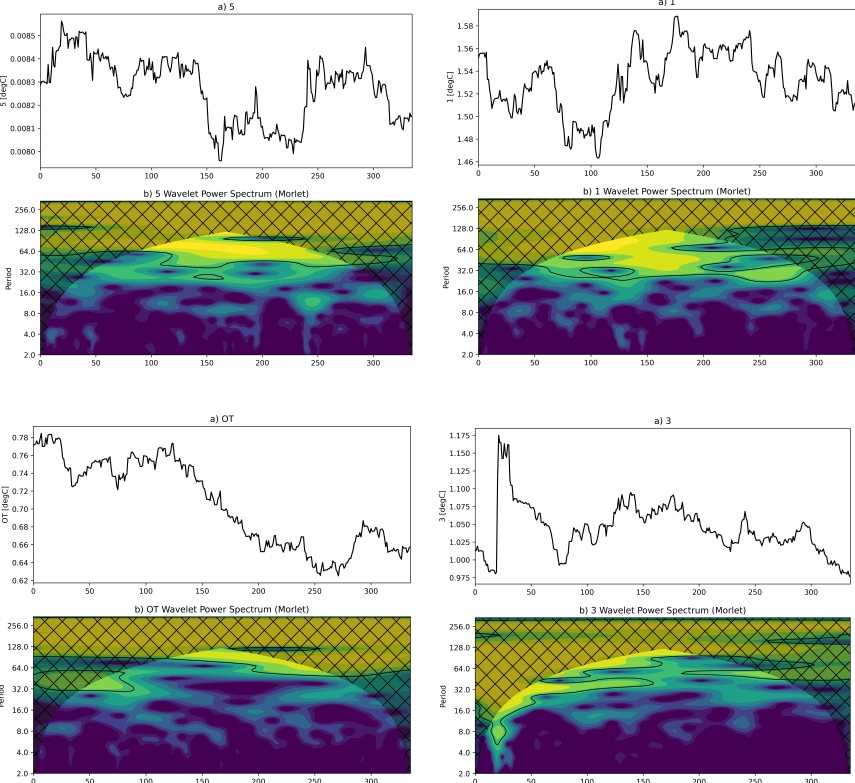

Figure 16: Exchange dataset

