# OpenReview forum: "TwinS: Revisiting Non-Stationarity in Multivariate Time Series Forecasting"
_ICLR.cc/2024/Conference — Submitted to ICLR 2024_

### Official Review · Reviewer_xxAP · 2023-10-20

**Soundness:** 2 fair
**Presentation:** 1 poor
**Contribution:** 1 poor
**Rating:** 3
**Confidence:** 4

**Summary:**

The paper addresses the challenges of non-stationarity in multivariate time series forecasting. The authors propose the TwinS model, which consists of three modules: Wavelet Convolution, Period-Aware Attention, and Channel-Temporal Mixed MLP. The Wavelet Convolution handles nested periodicity for time series embedding, the Period-Aware Attention models the periodic distributions at refined receptive fields, and the Channel-Temporal Mixed MLP learns the representation within time series patches across different variates. TwinS achieves state-of-the-art performance on eight datasets compared to mainstream time series models.

**Strengths:**

1. The paper offers insights by exploring the time-frequency representations of time series through wavelet analysis.
2. Detailed ablations and alternative implementations are provided to validate the effectiveness of each proposed component in the model.
3. The model achieves promising performance on eight datasets compared to recent time series models.

**Weaknesses:**

1. The paper is not well-written, especially because some key concepts are not elaborated, such as periodic missing states, hysteresis, and non-stationary periodic distribution. Besides, some abbreviations are not informal and inappropriately used (dim, Conv, TS, TSformers), and the figures are not portable. Thus, this paper essentially needs further polishing.
2. The method part can be difficult to follow. In the last paragraph of Section 3.1, I still don't know how the input is embedded into the outputted representation because it is almost separated from the previous paragraph only introducing the relationship between transforms and convolutional layers. I hope the authors can elaborate more on it.
3. The proposed framework somewhat lacks novelty. Both wavelet analysis and patching by windows are previously used techniques for analyzing time series. The author does not compare other modular designs of related work like FEDformer. The overall architecture seems a refinement of Crossformer (What is the difference between Channel-Temporal Mixed MLP and FFN in Crossformer?) but endowed with a higher computational cost.
4. Unclear motivations. As non-stationarity is commonly reflected by changing distribution statistics, in particular, its mean, variance, or temporal covariance varies over time. This paper focuses more on the changing periodic distribution issues, which are not well-acknowledged to stem from non-stationarity but the inherent properties of the time series itself. Could the author give more examples of the periodic distribution in highly non-stationary series and relatively stationary series?
5. About the experiment parts. The performance improvement of non-stationary forecasting seems incremental. And the model can also perform worse than previous models on highly non-stationary data (Exchange). Also, the authors offer two variants of the model, but only solely one of the proposed models does not yield consistent promising results, indicating that model selection is inevitable in practical applications. Besides, The analysis of the model is not systematic enough. Some experiments (Tables 2,3,4) only use the results on partial datasets.  It seems there is cherry-picking in particular configurations. Could the authors provide more complete results?
6. I find the code implementation is incorporated with re-normalization, Channel-Independence, and Patchfy strategies but are not shown in Fig.2 and the method part. Since each of them can work as a general performance booster, how much do these techniques help the model, and is it dominant?

**Questions:**

1. About the efficiency of the proposed method. Could you provide some memory/inference time evaluations?
2. How to explain the performance fluctuations by the choices of hidden dimensions?
3. How the Channel-Temporal Mixer Module capture the overall relations among time series as stated in the abstract?

---

### Official Review · Reviewer_iQeh · 2023-10-27

**Soundness:** 3 good
**Presentation:** 3 good
**Contribution:** 2 fair
**Rating:** 5
**Confidence:** 4

**Summary:**

This paper proposes a Transformer-based for handling the non-stationary time series, especially with characteristics of diverse periodic patterns. The proposed model consists of the Wavelet Convolution, Period-Aware Attention, and Channel-Temporal Mixed MLP. The experimental results verified its effectiveness.

**Strengths:**

1. This work is easy to follow.

2. The motivation is clear.

3. The experimental results seem to be convincing.

**Weaknesses:**

1. While the motivations and introduction of the proposed method are quite clear, there are still areas that could be refined. For example, which component contributes to handling periodic distributions in non-stationary time series forecasting? I would appreciate more clarifications on the effects of each component of the proposed model.

2. The discovery of periodic distributions, including multiple nested and overlapping periods, in non-stationary time series is not a novel concept. From a theoretical standpoint, Kuznetsov, Vitaly, and Mehryar Mohri [1] have already established generalization bounds by leveraging periodic distributions. On a practical note, Zhang and Zhou [2] employed a two-step approach for prediction, selectively ensembling periodic patterns. These prior studies, along with their variations, should be discussed or even compared within the context of this paper.

Certainly, with the progression of time and technology, there's an intuitive inclination to employ large models to tackle complex problems. However, this doesn't imply that we should resort to a completely black-box or consequential approach. It's crucial to comprehend why the traditional transformer struggles with non-stationary data or what specific limitations arise when handling the non-stationary data outlined in this article. These points should be clearly delineated in Figure 1.


[1] Kuznetsov, Vitaly, and Mehryar Mohri  "Learning theory and algorithms for forecasting non-stationary time series." NIPS'2015.
[2] Zhang, Shao-Qun, and Zhi-Hua Zhou. "Harmonic recurrent process for time series forecasting." ECAI'2020.

**Questions:**

As mentioned in Weaknesses. I would raise my score if the authors fixed my doubts in the rebuttal.

---

### Official Review · Reviewer_j2XZ · 2023-10-31

**Soundness:** 2 fair
**Presentation:** 2 fair
**Contribution:** 2 fair
**Rating:** 5
**Confidence:** 4

**Summary:**

In this paper, the authors investigate the problem of non-stationary time-series forecasting. Since the non-stationary time series data are charactered with multiple-nested and overlapping periods, distinct periodic pattern and the significant hysteresis, the authors propose the TWINS, which contains the wavelet convolution, period-aware attention and the channel-temporal mixed MLP.

**Strengths:**

1.	The authors bridge the relationship between the Gabor transform and the convolution neural networks in time-series data.
2.	The authors evaluate the proposed methods on several datasets and achieve ideal performance.

**Weaknesses:**

1.	The motivation is not clear. It is hard to find the period information in Figure 1. Specifically, there are no nested and overlapping periods as well as hysteresis in the example the authors provided. Moreover, it is suggested that the authors should introduce the meaning of the red frame in Figure 1.
2.	In challenge (ii), the authors claim that the traditional attention mechanisms primarily rely on explicitly modeling period information through the values of each time step, but they do not mention any citation and provide any evidence, which is not persuasive.
3.	The authors employ the Wave transform, which is one type of spectral transformation in time-series processing. There are several methods that address the time-series forecasting problem with the technique of spectral transformation, for example, fast Fourier Transform. What is the advantage of Wave transform in nonstationary forecasting compared with other spectral transformation methods?
4.	As for the discussion of non-stationary time series methods in related works, the authors mentioned that some methods focus on evolving statistical features and overlook periodic distributions. Please justify the difference between these statistical features and periodic distributions. Can the author provide any concrete examples?
5.	On page 5, the authors propose the Periods Detection Attention, which is used to model the nonstationary periodic. But this is heuristic. It is suggested that the authors should provide and visualization to prove that this mechanism can identify the periodic information.
6.	It is suggested that the authors should consider more compared methods for nonstationary time-series forecasting like[1].

[1] Koopa: Learning Non-stationary Time Series Dynamics with Koopman Predictors

**Questions:**

Please refer to Weakness

---

### Official Review · Reviewer_i7Bk · 2023-10-31

**Soundness:** 3 good
**Presentation:** 1 poor
**Contribution:** 2 fair
**Rating:** 5
**Confidence:** 3

**Summary:**

This paper proposes TwinS, which consist of Wavelet Convolution, Period-Aware Attention and Channel-Temporal Mixed MLP to solve non-stationary problem in time-series forecasting.

**Strengths:**

1. This paper puts forward a novel perspective on the periodicity of time series non-stationarity, breaking through the limitations of some previous methods that only looked at non-stationarity from statistical information in the time domain.
2. This paper proposes Wavelet Convolution, Period-Aware Attention and Channel-Temporal Mixed MLP to solve nested periodicity, absence of periodic distributions and hysteresis among the variable, respectively.
3. The experimental results show promising performance of TwinS.

**Weaknesses:**

1. Many design details of network structure are not clearly described.
2. Although the motivation of each part of the network is described clear, many designs do not match with the motivations well.


See Questions.

**Questions:**

1. In eq. (8), what is window rotation operation and why is can improve the model's ability to resist outliers.
2. In eq. (11), is $W_p$ a learnable parameter? If so, then the $W_{score}^{(ls)}$ is not an attention score as it does not include interactions between patches.
3. How is eq. (11) "periodic Aware"? How do you make sure that these head captures dependencies in different frequencies as shown in Figure 4.
4. What does "align the number of detection heads and attention heads" mean? How are $W_{score}^{(ls)}$s aggregated to $W_{score}^{(lm)}$?
5. In eq. (16), the channel-temporal mixer MLP does not seem to mix any thing as it is only as two-layer MLP applied to a D' size tensor.

---

### Meta-Review · Area_Chair_vet1 · 2023-12-10

**Metareview:**

Reviewers reached unanimous negative opinion on this paper. While the paper has some merits in studying the non-stationarity of time series using some new techniques such as Wavelet, since no author rebuttal was provided, the open issues remained unresolved.

**Justification For Why Not Higher Score:**

All reviewers recommended rejection. No author rebuttal.

**Justification For Why Not Lower Score:**

N/A

---

### Decision · Program_Chairs · 2024-01-16

Reject